# The Tyrosine Phosphatase Activity of PTPN22 Is Involved in T Cell Development via the Regulation of TCR Expression

**DOI:** 10.3390/ijms241914505

**Published:** 2023-09-25

**Authors:** Bin Bai, Tong Li, Jiahui Zhao, Yanjiao Zhao, Xiaonan Zhang, Tao Wang, Na Zhang, Xipeng Wang, Xinlei Ba, Jialin Xu, Yang Yu, Bing Wang

**Affiliations:** Key Laboratory of Bioresource Research and Development of Liaoning Province, College of Life Science and Health, Northeastern University, #195 Chuangxin Road, Hunnan Xinqu, Shenyang 110169, China; liuyusheng34@163.com (B.B.); 2101394@stu.neu.edu.cn (T.L.); jiahuizhao1130@163.com (J.Z.); zhaoyanjiao1996@126.com (Y.Z.); zhangxn@bbmc.edu.cn (X.Z.); 1510061@stu.neu.edu.cn (T.W.); 1910070@stu.neu.edu.cn (N.Z.); m13674205861@163.com (X.W.); 2001355@stu.neu.edu.cn (X.B.); jialin_xu@mail.neu.edu.cn (J.X.)

**Keywords:** PTPN22, TCR, T cell development, internalization, recycling

## Abstract

The protein tyrosine phosphatase PTPN22 inhibits T cell activation by dephosphorylating some essential proteins in the T cell receptor (TCR)-mediated signaling pathway, such as the lymphocyte-specific protein tyrosine kinase (Lck), Src family tyrosine kinases Fyn, and the phosphorylation levels of Zeta-chain-associated protein kinase-70 (ZAP70). For the first time, we have successfully produced PTPN22 CS transgenic mice in which the tyrosine phosphatase activity of PTPN22 is suppressed. Notably, the number of thymocytes in the PTPN22 CS mice was significantly reduced, and the expression of cytokines in the spleen and lymph nodes was changed significantly. Furthermore, PTPN22 CS facilitated the positive and negative selection of developing thymocytes, increased the expression of the TCRαβ-CD3 complex on the thymus cell surface, and regulated their internalization and recycling. ZAP70, Lck, Phospholipase C gamma1(PLCγ1), and other proteins were observed to be reduced in PTPN22 CS mouse thymocytes. In summary, PTPN22 regulates TCR internalization and recycling via the modulation of the TCR signaling pathway and affects TCR expression on the T cell surface to regulate negative and positive selection. PTPN22 affected the development of the thymus, spleen, lymph nodes, and other peripheral immune organs in mice. Our study demonstrated that PTPN22 plays a crucial role in T cell development and provides a theoretical basis for immune system construction.

## 1. Introduction

Thymocytes undergo multiple processes in order to develop into mature T cells. Initially, CD4^−^CD8^−^ DN (double negative) cells were divided into four stages: DN1 (CD25^−^CD44^+^), DN2 (CD25^+^CD44^+^), DN3 (CD25^+^CD44^−^), and DN4 (CD25^−^CD44^−^) [1,2]. The TCRβ and pTα chains cooperate to promote the development of DN cells into CD4^+^CD8^+^ DP cells and rearrange the TCRα chain to express mature α/β TCR (T cell receptor) [3]. Double positive (DP) T cells differentiate into CD4^+^ or CD8^+^ SP (single positive) T cells through positive and negative selection [4,5]. T cell development and activation are modulated by TCR. Both positive and negative selection relies on the affinity for self-pMHC (cognate peptide-MHC) and the signal from the TCRαβ-CD3 complex [6,7]. 

TCR binds to the pMHC on antigen-presenting cells (APCs), leading to the activation of C-terminal Src kinase (Src) family proteins such as Lck and Fyn [8,9]. CD4 and CD8 enhance TCR-recruiting Lck; however, high-affinity ligands induce the signals independently [6]. Activated Lck phosphorylates the ζ chain to transfer the signal into the cell interior. Subsequently, ZAP70 activates the linker of activated T cells (LAT), PLCγ1, and the Gads/SLP-76/ITK complex. The MAPK/ERK pathway and calcium mobilization are crucial for the survival, development, and differentiation of thymocytes [10,11]. Consequently, the expression of TCR on the surface of T cells is essential for thymocyte development.

The TCRαβ-CD3 complex consists of α chain, β chain, and CD3. The balance of TCR expression on the cell surface depends on its production, vesicle trafficking, internalization, recycling, and degradation. Because production and degradation occur at a slow pace, internalization and recycling are the principal mechanisms that maintain the balance [12]. TCR expression levels on the T cell surface are a dynamic equilibrium [13]. When T cells are activated by TCR, immune synapses are formed between the T cell surface and the APCs. Internalization, recycling, and degradation of TCR play essential roles in the signaling transmission between APCs and T cells [14]. However, the membrane trafficking of TCRs, which is crucial for sustaining the TCR signal, is still not fully understood.

PTPN22, also known as proline-enriched phosphatase (PEP), has been found to interact with the Src homology 3 (SH3) domain of C-terminal Src kinase (Csk) [15]. The PTPN22 human homolog-lymphoid phosphatase (Lyp) is highly expressed in the thymus and spleen [16]. PTPN22 dephosphorylates TCR signaling proteins, including Lck, Fyn, CD3ζ, and ZAP70, which have been identified as their substrates [17,18]. PTPN22 interacts with Csk to dephosphorylate Lck Y394 and inhibit the activity of Src kinases. Recently, we have identified EB1 as a new partner associated with PTPN22 in the regulation of TCR signaling and T cell immune responses [19]. Taken together, these studies indicate that PTPN22 acts as a negative regulator of the TCR signaling pathway. 

Lyp-R620W (Lyp is an expression form of PTPN22 in the human body) cannot interact with Csk, resulting in autoimmune diseases such as rheumatoid arthritis, type 1 diabetes, and systemic lupus erythematosus [20,21]. PTPN22 R619W transgenic mice display thymic and splenic enlargement and hyper-responsiveness of lymphocyte and dendritic cells [22]. However, no statistically significant disparity was observed in thymocyte numbers or subsets in PTPN22 knockout mice until they reached the age of six months [23]. Until now, studies of the biological function of PTPN22 have relied primarily on these two mouse models. Researchers mutated Cys227 to Ser (C227S) to inhibit PTPN22 phosphatase activity [18] and generate transgenic mice expressing PTPN22 C227S in order to investigate the impact of PTPN22 enzyme activity on the initial development of the immune system in mice. 

In our study, when the phosphatase activity of PTPN22 in mouse thymocytes was inhibited, the number of thymocytes decreased. PTPN22 CS increased the expression of TCRβ^med/hi^ and CD3 ^med/hi^ on the thymocyte surface and promoted positive and negative selection. In addition, it regulated the internalization and recycling of TCR. In PTPN22 CS mouse thymocytes, cell proliferation ability was increased mainly through the upstream TCR signaling pathway, and the phosphorylation of TCR signal pathway proteins such as ZAP70, Lck, LAT, and PLCγ1 was also increased. Of these proteins, PLCγ1 is crucial for vesicle transport. These results show that PTPN22 regulates internalization and recycling via the TCR signaling pathway. Overexpressed TCRαβ-CD3 on the T cell surface promoted the positive and negative selection of T cells, and the development of T cells was promoted. Our study provides a theoretical basis for understanding the relationship between the enzyme activity of PTPN22 and autoimmune diseases.

## 2. Results

### 2.1. Production of the PTPN22 CS Transgenic Mice 

PTPN22 knockout mice showed no apparent differences in numbers of thymocytes or subsets before six months of age [23]. We diluted the endogenous PTPN22 by overexpressing PTPN22 CS to inhibit the phosphatase activity of PTPN22. As a result, PTPN22 CS transgenic mice were created in order to investigate the function of PTPN22 in T-cell development. Use of the PTPN22 CS transgenic mice avoided the compensatory effects of proline-, glutamic-, serine- and threonine-rich (PEST) or PTPN18 in PTPN22 knockout mice. Using CRISPR-Cas9 technology, gRNAs were coinjected with the CAG promoter-loxP-3* polyA-loxP-Kozak-mutant *Ptpn22* CDS (p. C227s, TGC to AGC)-PolyA into fertilized mouse eggs to generate *Ptpn22 CS* gene-positive mice. Lck Cre mice contain a proximal *Lck*-specific promoter. PTPN22 CS homozygous flox mice (flox/flox) were mated with Lck Cre mice to obtain specific overexpression of PTPN22 CS in the thymus (Figure 1A). The PTPN22 CS flox/flox genotype mice were unable to express the exogenous PTPN22 CS protein and these were identified as control mice. The PTPN22 CS flox/flox-Lck Cre genotype mice specifically expressed PTPN22 CS protein in thymus cells, and these were identified as PTPN22 CS mice. The bred mice were identified by flox and the Lck Cre gene (Figure 1B). In the PTPN22 CS mice, the expression of PTPN22 mRNA increased 6.3-fold, and protein increased 2.3-fold (Figure 1C,D). After development in the thymus, mature T cells emigrated to peripheral organs such as the spleen. There was no significant difference between the spleen cells of the control and PTPN22 CS mice spleen cells in terms of mRNA and protein expression (Figure 1E,F). These results demonstrated the effectiveness of Lck in constraining the specific expression of PTPN22 CS in the mouse thymus. In addition, we used RT-PCR and ELISA to test the expression of cytokines in the spleen and lymph nodes. The expression of IL-2, IL-7, and IFNγ were increased in PTPN22 CS mice (Figure 1H,I). It has been proposed that the activation and immune response of peripheral organ cells is regulated by PTPN22 CS in the thymus. These results indicate that we were successful in producing PTPN22 CS transgenic mice. 

We analyzed more than 500 pairs of four-to six-week-old mice and detected their thymocytes using flow cytometry. The average numbers of thymus cells in the control and PTPN22 CS mice were 2.86 × 10^7^ and 1.93 × 10^7^ respectively (Figure 1G). These results suggested that PTPN22 CS regulates the development of mouse thymocytes.

### 2.2. PTPN22 CS Promotes the Development of DN Cells

In the thymus, the development of pre-T cells starts in the CD4^−^CD8^−^ DN cells. Therefore, we analyzed the function of PTPN22 CS in the development of DN cells. The DN1, DN2, DN3, and DN4 phases were distinguished by CD44^+^CD25^−^, CD44^+^CD25^+^, CD44^−^CD25^+^, and CD44^−^CD25^−^. We analyzed CD3^−^CD4^−^CD8^−^ triple negative thymocytes using flow cytometry. PTPN22 CS reduced the expression of CD25 and CD44 (Figure 2A,B). The proportions of DN1 cells in the control and PTPN22 CS mice were similar. DN2 and DN3 cell proportions were reduced in the PTPN22 CS mice, whereas the proportion of DN4 cells was increased significantly (Figure 2C). This indicated that PTPN22 CS promotes the development of cells from the early stage of double-negative cells to the DN4 stage. These results suggested that PTPN22 CS promotes the development of DN cells.

### 2.3. PTPN22 CS Promotes the Development of DP Cells

DP cells are divided into four stages according to the expression of CD69, CD5, and TCRβ. In stage 1, DP cells express TCRβ^lo^, CD69^lo^, and CD5^lo^; in stage 2, they express TCRβ^lo/med^, CD69^med/hi^, and CD5^med^; in stage 3, they express TCRβ^med/hi^, CD69^hi^, and CD5^hi^ and in stage 4, they express TCRβ^hi^, CD69^lo^, and CD5^hi^. Our results showed that PTPN22 CS decreased the proportions of TCRβ^lo^CD69^lo^ and TCRβ^lo^CD5^lo^ cells but increased the proportions of TCRβ^lo/med^CD69^med/hi^, TCRβ^med/hi^CD69^hi^, TCRβ^hi^CD69^lo^, TCRβ^lo/med^CD5^int^, TCRβ^med/hi^CD5^hi^, and TCRβ^hi^CD5^hi^ cells (Figure 3). These results suggested that the development of cells from DP stage 1 to stages 2, 3, and 4 was facilitated by PTPN22 CS. We also compared the MFI of TCRβ^lo^, TCRβ^med^, and TCRβ^hi^ in the PTPN22 CS and control mice. These results showed that TCRβ^med^ and TCRβ^hi^ increased in the PTPN22 CS mice. In summary, PTPN22 CS promoted DP cell development.

### 2.4. PTPN22 CS Promotes the Expression of TCRβ^med/hi^ and CD3^med/hi^ on T Cell Surface

DP cells develop from DN cells. TCRα, TCRβ, and CD3 combine into the TCRαβ-CD3 complex to be recognized by MHC. Compared with the control mice, the expressions of TCRβ^med/hi^, TCRα^med/hi,^ and CD3^med/hi^ on the surface of PTPN22 CS mouse thymus cells were significantly higher (Figure 4A–C). These results suggested that PTPN22 CS may affect T cell development by increasing the expression of the TCRαβ-CD3 complex on the cell surface. 

DN, DP, and SP cells can be distinguished based on the expression of CD4 and CD8. We therefore detected the expression of CD4 and CD8 on the surface of mouse thymus cells. Our results showed that the expression of CD4 in the thymus of PTPN22 CS mice was greater than in those of the control mice, and the expression of CD8 was lower than in those of the control mice (Figure 4D,E). In addition, PTPN22 CS affected the proportions of DN, DP, and SP cells in the mouse thymus. In particular, PTPN22 CS reduced the proportion of DN cells (Figure 4F). Our results showed that PTPN22 CS affects DP cell development by regulating the expression of the TCRαβ-CD3 complex on the cell surface. 

### 2.5. PTPN22 CS Promotes the Positive and Negative Selection of T Cells

DP cells develop into mature CD4 or CD8 SP T cells via positive and negative selection. Therefore, in order to analyze the regulatory effect of PTPN22 on T cell development, it is important to investigate the relationship between PTPN22 CS and positive/negative selection. We, therefore, bred PTPN22 CS mice with H-Y TCR-transgenic mice to obtain progeny expressing both H-Y and PTPN22 CS. H-Y TCRs are expressed in H-Y transgenic mice and mediate the negative selection of cells by binding to H-Y antigens. H-Y is a male-specific minor histocompatibility antigen. Therefore, H-Y TCR cells are negatively selected because of the presence of the H-Y antigen in male mice. H-Y TCR cells were positively selected in the context of H-2^b^ molecules in female mice [24,25]. 

The number of thymocytes was analyzed using flow cytometry. The average numbers of H-Y-control male (H-Y-control-M) and H-Y-PTPN22 CS male (H-Y-PTPN22 CS-M) mouse thymus cells were 2.26 × 10^7^ and 1.54 × 10^7^, respectively (Figure 5A). Compared with H-Y-control-M mice, the expressions of exogenous α chain (anti-T3.70), exogenous β chain (anti-F23.1), and total β chain (anti-H57) on the thymus cell surface of H-Y-PTPN22 CS-M mice were lower (Figure 5B). In addition, the proportion of DP cells in the thymus of H-Y-PTPN22 CS-M mice was increased, whereas the proportion of DN cells was decreased (Figure 5C). These results suggested that PTPN22 CS leads to premature apoptosis of thymocytes by promoting cells to be negatively selected. Surviving cells expressed fewer exogenous TCRαβ. Cells with few exogenous TCRαβ complexes could not be positively or negatively selected. A large number of cells with low expression of α chain and TCRβ chain accumulated in the DP stage, resulting in an increased proportion of DP cells and a decreased proportion of DN cells. T cells with exogenous TCRαβ cannot be negatively selected in female mice, and positive selection can be promoted to increase the proportion of SP cells. The average numbers of thymus cells in the H-Y-control female (H-Y-control-F) and H-Y-PTPN22 CS female (H-Y-PTPN22 CS-F) mice were 2.68 × 10^7^ and 1.85 × 10^7^, respectively. Compared with the H-Y-control-F mice, more cells expressed exogenous β chain in the H-Y-PTPN22 CS female mice, and the proportions of CD4^+^ SP cells were higher (Figure 5E,F), indicating that PTPN22 CS promotes positive selection. Overall, PTPN22 CS promotes both positive and negative selection by increasing the expression of TCR on the cell surface and modulating the proportion of DP and SP cells.

### 2.6. PTPN22 CS Regulates TCR Internalization and Recycling

PTPN22 CS modulates both positive and negative selection by affecting TCR expression on the surface of T cells. CD3 is an important part of the TCR complex. In this regard, we investigated CD3 expression on the cell surface at different stages of T cell development. The expression of CD3^med/hi^ was higher on the surface of CD4^+^CD8^−^ and CD4^+^CD8^+^ cells in the PTPN22 CS mice compared to the control mice. However, PTPN22 CS had less effect on the expression of CD3 on the surface of CD4^−^CD8^+^ and CD4^-^CD8^-^ cells (Figure 6A). These results indicated that PTPN22 CS increased the expression of CD3 on the surface of CD4^+^ cells but not CD4^−^ cells. We sorted CD4^+^ and CD4^-^ cells to analyze the CD3 expression in thymocytes. We found that although PTPN22 CS promoted the expression of CD3 on the surface of CD4^+^ cells, it did not affect the protein expression of CD3 in the interior of CD4^+^ or CD4^−^ cells (Figure 6B,C). We treated the thymocytes with CHX to inhibit protein synthesis and then analyzed the regulation by PTPN22 CS of the CD3 degradation rate in thymus cells. Compared with the control mice, the degradation rate of CD3 in PTPN22 CS mice thymus cells did not change significantly (Figure 6D). These results suggested that PTPN22 CS does not regulate CD3 expression on the surface of T cells by affecting CD3 degradation.

After excluding the effect of total CD3 protein expression and degradation on CD3 expression on the T cell surface, we analyzed whether PTPN22 CS regulates TCR expression on the T cell surface by affecting cell internalization or recycling. On the cell surface, CD3 was labeled with fluorescent antibodies. Flow cytometry was used to analyze the MFI of cells at different time points. When CD3 on the cell surface was internalized into the cell interior, the MFI would be lower. In our results, the internalization of PTPN22 CS mouse T cells was inhibited (Figure 6E). The bound fluorescent antibodies were washed from the cell surface and the fluorescent antibodies were returned to the cell surface by recycling. We used flow cytometry to detect the MFI in order to analyze the recycling of T cells. In the PTPN22 CS mouse T cells, the recycling of TCR was promoted (Figure 6F). These results showed that PTPN22 CS increased the expression of CD3 on the T cell surface by inhibiting the internalization and promoting the recycling of TCR.

### 2.7. PTPN22 CS Regulates T Cell Development Depending on the Upstream TCR Signaling Pathway

Cells were treated with CD3/CD28 or PMA + Ionomycin to analyze the effect of PTPN22 CS in different signaling pathways. When the cells were stimulated by CD3/CD28, the proliferation of thymus cells in PTPN22 CS mice was significantly improved in comparison with the control mice (Figure 7A). When PMA + Ionomycin was used to stimulate the cells, PTPN22 CS had less effect on the proliferation of mouse thymocytes (Figure 7B). Our results showed that PTPN22 CS regulated the activation, proliferation, and development of T cells primarily through the upstream TCR signaling pathway. The activity of many essential proteins in the TCR signaling pathway changes according to their own phosphorylation. We therefore examined the phosphorylation of these proteins in PTPN22 CS and control mice using Western blotting. CD3/CD28 increased the phosphorylation levels of ZAP70, PLCγ1, Lck, and Erk. When cells were stimulated by CD3/CD28 for the same length of time, the phosphorylation levels of these proteins in the thymus cells of the PTPN22 CS mice were significantly higher than in those of the control mice (Figure 7C). Of these proteins, PLCγ1 regulates vesicle transport.

Our results showed that PTPN22 CS improved the phosphorylation of essential proteins in the TCR signaling pathway. It affected the internalization and recycling of T cells by promoting the activation of the TCR signal pathway, promoting the expression of the TCRαβ–CD3 complex on the T cell surface, and regulating the development of T cells. In addition, PTPN22 CS facilitated T cell development to regulate the development of the thymus and affected the development of the spleen, lymph, and other peripheral organs.

## 3. Discussion

Protein tyrosine phosphatase PTPN22 is exclusively expressed in hematopoietic cells. It binds to Csk and negatively regulates TCR signaling by dephosphorylating Lck at Tyr394 [15]. Several PTPN22 substrates have been identified, including ZAP70, CD3ε, CD3ζ, and VAV1 [18]. PTPN22 inhibits the phosphorylation of TCR-CD3ζ in the primary TCR signal pathway [26]. However, no report has shown whether PTPN22 directly influences the expression of the TCRαβ-CD3 complex on the T cell surface or regulates T cell development. In our study, we identified a new role for PTPN22 in the regulation of TCRαβ–CD3 complex expression and T cell development. 

The weak phenotype of PTPN22^−/−^ mice may result from the functional replacement by two other subfamily members of PTP-PEST and PTP-HSCF (named PTPN18 in mice) [23,27,28]. Therefore, we generated transgenic mice with the overexpression of PTPN22 C227S mutation (PTPN22 CS), in which the loss of phosphatase activity simulated the lack of PTPN22 function. PTPN22 CS mice avoid compensation effects and amplify phenotypic differences. Although wild-type PTPN22 tyrosine phosphatase activity is still present in PTPN22 CS mice, the overexpression of PTPN22 CS without phosphatase activity significantly dilutes the PTPN22 concentrations and inhibits the dephosphorylation ability of endogenous PTPN22. 

T cells develop from mesenchymal stem cells, emigrate from the thymus, travel to the spleen and other organs, and enter the bloodstream. The proximal Lck promoter restricts the specific expression of exogenous PTPN22 CS in the mouse thymus [29,30]. By detecting mRNA and proteins in mouse thymus and spleen cells, we demonstrated that PTPN22 CS is specifically expressed in the PTPN22 CS mouse thymus. Mature T cells exercise their immune capacity in peripheral organs such as the spleen and lymph nodes, for example, by participating in immune responses. In the spleen and lymph nodes of PTPN22 CS mice, the expressions of IL-2, IL-7, and IFNγ were significantly increased (Figure 1H,I). These results suggest that PTPN22 CS promotes the expression of IL-2, IL-7, IFNγ, and other cytokines in peripheral organs by regulating the development of thymus T cells. In addition, we found that PTPN22 CS increased the proportion of mature CD4^+^ SP T cells in the thymus (Figure 4F). The strong signal from TCR leads to the development of DP cells into CD4^+^ SP T cells [31]. As T helper cells, CD4^+^ SP cells mainly secrete IL-2, IFNγ, and other cytokines [32,33,34]. We demonstrated that the proportions of CD4^+^ and CD8^+^ T cells in spleen cells were increased and decreased, respectively (Appendix A). ​It is possible that increasing the TCR expression enhanced the TCR signal. In summary, we successfully produced PTPN22 CS transgenic mice without PTPN22 phosphatase activity and demonstrated that overexpression of PTPN22 CS in the mouse thymus modulates the composition of immune cells in the peripheral organs by influencing T cell development.

PTPN22 CS reduced the expression of CD25 and CD44 on the thymocyte surface. The percentages of DN2 and DN3 cells were decreased, and the percentage of DN4 cells was significantly increased (Figure 2C). These results suggested that PTPN22 CS enhances the development of DN2 and DN3 cells development into DN4 cells. Furthermore, the proportion of DN cells in the thymus of PTPN22 CS mice was reduced (Figure 4F). This indicated that PTPN22 without phosphatase activity promoted the development of DN cells into DP cells. Mature T cells emigrate from the thymus and travel to peripheral immune organs, and it was therefore not sufficient to prove whether PTPN22 CS promoted or inhibited thymus cell development. H-Y mice were introduced to further analyze the effect of PTPN22 CS on positive and negative selection.

There were 86.1% DP and 2.9% DN cells in the thymus of the control mice, compared with 8.4% DP and 78.7% DN cells in the H-Y-control-M mice (Figure 4F and Figure 5C). The H-Y TCRs are encoded in H-Y transgenic mice and bind to the H-Y antigen that is expressed in male mice; these cells will be subject to apoptosis [25,35]. Therefore, during the development of the H-Y-control-M mouse thymus, a large number of cells that expressed exogenous H-Y TCRαβ were negatively selected. In H-Y-PTPN22 CS-M mice, PTPN22 CS promoted the development of DN cells into DP cells and expression of TCRαβ on the cell surface. Increased expression of TCRαβ on the cell surface promoted negative selection of these cells via binding to the H-Y antigen, thereby reducing the number of thymus cells in H-Y PTPN22 CS-M mice (Figure 5A). The residual cells expressed few TCRαβ, so they were blocked from positive or negative selection. Finally, the cells with low expression of TCRαβ on their surface remained in the DP stage, resulting in an increase in the proportion of DP cells and a decrease in the proportion of DN cells. These results suggested that PTPN22 CS promoted negative selection. In the H-Y transgenic female mice, there was no male-specific H-Y antigen and apoptosis could not be mediated by H-Y TCR, but the percentage of SP cells was increased by positive selection. Mature SP T cells are transported to peripheral immune organs. As the number of thymocytes in H-Y PTPN22 CS-F mice decreased, we suggest that PTPN22 CS promoted the positive selection of H-Y transgenic cells in female mice and that mature SP T cells were transported to peripheral immune organs. Therefore, the expression of TCRβ on the H-Y-PTPN22 CS-F mouse thymocyte surface was elevated (Figure 5E). These results show that PTPN22 CS promotes both positive and negative selection by increasing TCRαβ expression on the cell surface. These conclusions explained why there were fewer thymus cells in the PTPN22 CS mice than in the control mice (Figure 1G). PTPN22 CS increased the expression of TCRαβ–CD3 complexes on the surface of thymocytes to promote the cells to enter the positive or negative selection stages. Normally, about 5% of thymus cells enter the selection stage, and 85-90% of them are apoptotic because of negative selection [36]. These differences in negative or positive selection accumulated in the thymus of PTPN22 CS mice, which showed a significant decrease in the number of thymus cells. 

PTPN22 CS promoted CD3 expression on the surface of CD4^+^ cells but not CD4^-^ cells (Figure 6A). These results indicate that the regulation of CD3 expression on the T cell surface by PTPN22 CS is a relatively rapid process. The synthesis of CD3 is slower than CD3 recycling. We therefore ruled out the theory that PTPN22 CS regulates the expression of CD3 on the surface of T cells by affecting CD3 protein synthesis. TCR expression on the T cell surface maintains dynamic balance and is closely related to internalization and recycling [13]. Cell surface TCRαβ-CD3 complexes enter the cytoplasm through internalization [12]. Some TCRs are degraded by lysosome or proteasome pathways, and others return to the T cell surface via recycling [37]. When the total quantities of CD3 in T cells are similar, we can exclude the effects of synthesis and degradation; PTPN22 may regulate CD3 expression on the T cell surface via internalization or recycling. Our results showed that PTPN22 CS lost its phosphatase activity, inhibited CD3ε internalization, and enhanced CD3ε recycling to the T cell surface, resulting in increased expression of TCRβ^med/hi^ and CD3ε^med/hi^ on the surface of the T cells. It is possible that PTPN22 CS regulates the expression of the TCRαβ–CD3 complex on the T cell surface by inhibiting TCR internalization and promoting TCR recycling. 

There are various signal pathways in the cell that form a complex network. We have analyzed the PTPN22 CS-dependent signal pathway by stimulating T cells with CD3/CD28 or PMA + Ionomycin. We found that PTPN22 CS had a greater effect on T cell proliferation with CD3/CD28 stimulation. Therefore, we speculate that PTPN22 CS regulates T cell proliferation or development through the upstream TCR signal pathway. We examined the phosphorylation of some essential TCR signal pathway proteins in thymocytes to further demonstrate the relationship between PTPN22 CS and the TCR signal pathway. The tyrosine phosphorylation levels of ZAP70, PLCγ1, and other proteins increased significantly, which indicates that they were activated by PTPN22 CS. Our results showed that PTPN22 CS promotes T cell activation and influences T cell development by regulating the essential protein phosphorylation in the TCR signal pathway. Furthermore, we noted that the regulation of vesicle trafficking is an important function of PLCγ1 [38,39,40], supporting the hypothesis that the effect of PTPN22 CS on the expression of the TCRαβ–CD3 complex on the thymocyte surface may be dependent on internalization and recycling.

In this study, we investigated a new function of PTPN22 in T cell development. When the phosphatase activity of PTPN22 is inhibited, the TCR internalization and recycling are regulated, depending on the TCR signaling pathway. Then, the expression of TCRβ^med/hi^ and CD3ε^med/hi^ is significantly increased in PTPN22 CS thymocytes. Overexpression of TCR promotes positive and negative selection and other developmental processes in T cells which subsequently affect the development of peripheral immune organs. Our study reveals a new role for PTPN22 without phosphatase activity in immune system homeostasis and provides new data that contribute to our understanding of autoimmune diseases. Our study demonstrates that PTPN22 influences T cell and immune organ development by regulating the expression of TCR on the surface of T cells and modulating the immune system. These results explained the relationship between PTPN22 and autoimmune diseases. When the phosphatase activity of PTPN22 is changed, the development of T cells is promoted, and the expression of TCR on the cell surface is increased, T cells will be more easily activated. Then, the T cells will have excess immune responses, which in turn increases the incidence of autoimmune diseases.

## 4. Materials and Methods

### 4.1. Mice

Using CRISPR-Cas9 technology, gRNA, “CAG promotor-Loxp-3 *polyA-loxP-Kozak-mutant *Ptpn22* CDS (p. C227S, TGC to AGC)-polyA”, and Cas9 mRNA were injected into fertilized mouse eggs to obtain positive F0 generation mice. The F0 generation mice were mated with wild-type (WT) mice, and PTPN22-CS heterozygous mice (flox/+) were selected from the offspring. PTPN22-CS homozygous flox mice (flox/flox) were obtained by self-crossing from F1 generation mice. Using the Cre-loxP system, PTPN22-CS homozygous flox mice (flox/flox) were intermated with Lck Cre mice, resulting in overexpression of PTPN22 CS protein in the thymus. Transgenic founders and the offspring were identified via tail genomic DNA using PCR (Appendix A). PTPN22 CS mice were bred with H-Y TCR transgenic mice (H-Y mice) from the Clinical Research Institute of Montreal (IRCM) and the progeny were backcrossed to generate H-Y-TCR PTPN22 CS transgenic mice. H-Y TCR expression was analyzed through flow cytometry using phycoerythrin-conjugated anti-H-Y TCR specific mAb (T3.70). All experiments were performed in accordance with protocols approved by the Laboratory Ethics Committee of Northeastern University and all animals were cultured in accordance with the Care and Use of Medical Laboratory Animals (National Institutes of Health, No. 8023).

### 4.2. Cell Culture

Thymocytes were isolated from control or PTPN22 CS mice. These cells were cultured in Roswell Park Memorial Institute (RPMI) 1640 Medium (Gibco, New York, NY, USA, 11875-093) and supplemented with 1% penicillin-streptomycin (Gibco, 15070063), 1% L-glutamine (Gibco, 25030149), and 10% FBS (Gibco, 10099). Cells were cultured in a 37 °C, 5% CO_2_ incubator.

### 4.3. Antibodies

Antibodies including anti-PTPN22 (#14693), anti-PLCγ1 (#5690), anti-GAPDH (#2118), anti-phospho-PLCγ1 (Y783) (#2821), anti-pErk1 (Thr202)/Erk2 (Tyr204) (#04370), anti-CD3ε (#4443) and anti-rat IgG antibody coupled to Alexa Fluor 488 (#4416) were purchased from Cell Signaling Technology (CST, Danvers, MA, USA). TCR H-Y (male antigen) monoclonal antibody (T3.70) was purchased from eBioscience (Thermo, Waltham, MA, USA) and anti-mouse TCR Vβ8 antibody (F23.1) was purchased from BD (Bio-Rad, Hercules, CA, USA). Anti-ZAP70 (ab32429), anti-pZAP70Tyr319 (ab131270), anti-Lck (ab3885), anti-pLckTyr394 (ab201567), and anti-Erk (ab17942) were purchased from Abcam (England). The anti-mouse CD3ε (16-0032-86) and anti-mouse CD28 (16-0281-85) antibodies were purchased from eBioscience. Mouse CD4 (L3T4) microbeads (130-049-201) were purchased from Miltenyi Biotec (Bergisch Gladbach, Germany).

### 4.4. Flow Cytometry

Cells were extracted from the thymus or spleen of the control or PTPN22 CS mice. Cells were incubated in 2% BSA for 10 min and mixed with fluorophore-conjugated antibodies for 40 min at 4 °C. The following antibodies used in flow cytometry were purchased from BD Pharmingen (Bio-Rad, Hercules, CA, USA): PE-CD3ε (561824), APC-CD3ε (553061), PerCP-CD4 (561090), APC-CD4 (561091), FITC-CD8 (561966), APC-CD8 (561093), PE-CD5 (553025), FITC-CD25 (561779) and PE-CD44 (561860). A quantity of 1~3 × 10^5^ cells was collected using LSRFortessa^TM^ (BD Pharmingen) and analyzed using Flow Jo software (Version 10). Antibody sources and dilutions are listed in Appendix A.

### 4.5. Proliferation Assays

Thymocytes from four- to six-week-old PTPN22 CS and control mice were stimulated using either phorbol-12-myristate-13-acetate (PMA) (50 ng/mL, Sigma Chemical Co, New Jersey, USA, P1585) and ionomycin (500 ng/mL, Beyotime Biotechnology (Shanghia, China), S1672), or CD3ε (5 μg/mL) and CD28 (1 μg/mL). They were cultured at 37 °C for 0, 24, 48, and 72 h. 3-(4,5-dimethylthiazol-2-yl)-5-(3-carboxymethoxyphenyl)-2-(4-sulfophenyl)-2H-tetrazolium (MTS) was added to the cells and they were then incubated for 4 h at 37 °C. Then, 25 μL of 10% SDS was added to the cells to stop the reaction, and the cells were detected at an optical density (OD) of 490. CellTiter 96^®^ AQueous One Solution Cell Proliferat (G3580) was purchased from Promega Corporation (Madison, WI, USA).

### 4.6. RNA Isolation and Quantitative Real-Time PCR (RT-PCR)

The RNA was purified using Total RNA Kit I (Omega, VT, USA, R6934). It was then reverse transcribed to cDNA using the GoScript™ Reverse Transcriptase Kit (Promega, Madison, WI, USA, A5003), and the gene expression levels were analyzed using GoTaq^®^ qPCR Master Mix (Promega, A6001) and reacted using the CFX Manager System (Bio-Rad Inc, Hercules, CA, USA). The amount of mRNA was calculated using a comparative thresholding method and expressed as a relative fold increase compared to the expression of the 18S gene. The primer sequences used for the RT-PCR analysis are listed in Appendix A.

### 4.7. Western Blotting

Thymus cells isolated from four- to six-week-old control or PTPN22 CS mice were stimulated using CD3 (5 μg/mL) and CD28 (1 μg/mL). Cells were cultured in RIPA (150 mM NaCl, 20 mM Tris/HCl pH = 7.4, 1 mM Na_3_VO_4,_ 5 mM EDTA containing 1% NP-40, 10 mg/mL aprotinin, leupeptin, and 1 mM PMSF, purchased from Sigma-Aldrich, NJ, USA) buffer at 4 °C for 30 min and then centrifuged at 13,000× *g* for 15 min. Proteins were separated by 10% SDS-PAGE gel and transferred to a PVDF membrane (Sigma-Aldrich, Inc. IPVH00010). PVDF membranes were washed in Tris-buffered saline with 0.5% Tween 20 (TBS-T) and blocked with 5% BSA-TBS-T at room temperature for 1 h. Protein was immunized overnight with primary antibodies. The protein signal was detected using ECL Western Blotting Detection Reagents (Sigma-Aldrich, Inc. WBULS0500). Blots were scanned using the ChemiDoc™ MP imaging system (Bio-Rad Laboratories, Hercules, CA, USA), and the results were analyzed using Image Lab™ Software 5.2.1 (Bio-Rad Laboratories).

### 4.8. Enzyme-Linked Immunosorbent Assay

Cells were obtained from the lymph nodes or spleen of the mouse. The blood cells were cleaned using the lysate of red blood cells. The samples were diluted into the range of concentrations using 1 × dilution buffer into the range of concentrations. IL-2, IL-7, and IFNγ expressions were analyzed using the ELISA kit from R&D Systems Minnesota, USA (DY485, DY402, and DY407). ELISA was performed in accordance with the instructions. Our experiments were performed in triplicate.

### 4.9. Internalization and Recycling Assays Using Flow Cytometry

A modified flow cytometric internalization assay was used to measure CD3ε internalization. Cells obtained from the thymus were resuspended in 100 μL of 2% BSA-PBS and then incubated on ice for 10 min. Cells (1 × 10^6^) were cultured in primary cell culture medium (RPMI 1640 containing 10% FBS, 2 mM glutamine, and 50 μM β-Mercaptoethanol, penicillin and streptomycin) and the cell surface CD3ε was labeled with anti-CD3ε (1 μg/mL) for 40 min at 4 °C. Samples were collected at different time points, 0, 15, 30, 45, 60, 90, and 120 min. The cells were washed twice with 1% BSA-PBS. The anti-CD3ε remaining on the cell surface was detected using the anti-rat IgG antibody together with Alexa Fluor 488 (CST, #4416). All samples were analyzed by flow cytometry, LSRFortessa^TM^ (BD Pharmingen) to detect Alexa Fluor 488 (FL1) fluorescence (mean fluorescence intensity, MFI). This is the calculation formula:$$\frac{Internalized}{Surface}\%=100\times \frac{\left(MFIt=0)-(\mathrm{MFIt}=n\right)}{\left(\mathrm{MFIt}=n\right)}$$

Recycling of CD3 was also analyzed using flow cytometry. A quantity of 1 × 10^7^ cells/mL thymocytes were cultured at 37 °C for 40 min in the primary cell culture medium containing 5 μg/mL PE-conjugated intact mouse CD3ε antibody. The cells were washed twice and then incubated at 37 °C for different time periods. The cells were cultured at 4 °C by ice-cold 1% BSA-PBS. Thymocytes were washed twice in 1% BSA-PBS (pH = 1.5) to remove the CD3ε antibody bound to recycled CD3ε. All these steps were performed away from the light. The thymocytes were analyzed by flow cytometry, LSRFortessa^TM^ (BD Pharmingen). This is the calculation formula:$$R\mathrm{ecycling}\%=100\times \frac{\left(MFIt=0)-(MFIt=n\right)}{(MFIt=0）}$$

### 4.10. Statistical Analysis

GraphPad Prism 8.0 was used for the statistical analysis. All of the data were acquired from at least three times independent experiments and presented as the mean ± SEM. The data were analyzed using an unpaired two-tailed Student’s t-test between the two groups. The differences of each group were calculated using a one-way analysis of variance (ANOVA), * *p* < 0.05, ** *p* < 0.01, *** *p* < 0.001, ns, not significant.

## 5. Conclusions

We have found that PTPN22 CS affects the development of immune organs such as the thymus, spleen, and lymph nodes. Therefore, we analyzed the regulation of PTPN22 on the immune system at the cellular and protein levels. PTPN22 without phosphatase activity in thymocytes affects TCR internalization and recycling by promoting the activation of the TCR signaling pathway. Therefore, the expression of TCRβ^med/hi^ and CD3 ^med/hi^ on the thymus cell surface is increased, and the positive and negative selection of the T cell development is promoted. This study reveals the novel role of PTPN22 in T cell development. In the future, we will look for vesicle transport proteins that are regulated by PTPN22, and explore the regulation mechanism between them. In addition, we need to combine basic research with clinical disease to explore how the effects of PTPN22 on T cell development regulate the pathogenesis of autoimmune diseases such as type 1 diabetes, systemic lupus erythematosus, and rheumatoid arthritis.

### Study Scheme

Experimental purpose: Study the regulation of PTPN22 CS enzyme activity on the development of mouse thymocytes.

Exploration process and experimental method (Figure 8): a)Constructed PTPN22 CS transgenic mice (CRISPR-Cas9 technology); verify the expression of protein and RNA of PTPN22 CS (Western blot, RT-PCR); detect the regulation of PTPN22 CS on thymus and peripheral immune organs (Flow cytometry; RT-PCR; ELISA).b)PTPN22 CS regulates T cell development. This process is defined by the expression of TCR on the surface of T cells (Flow cytometry).c)Study the causes of changes in TCR expression on the T cell surface (flow cytometry, Western blot).d)Study the signal pathway on which PTPN22 regulates the internalization and recycling of TCR (MTS, Western blot).

The flow chart is as follows:

**Figure 8 ijms-24-14505-f008:**
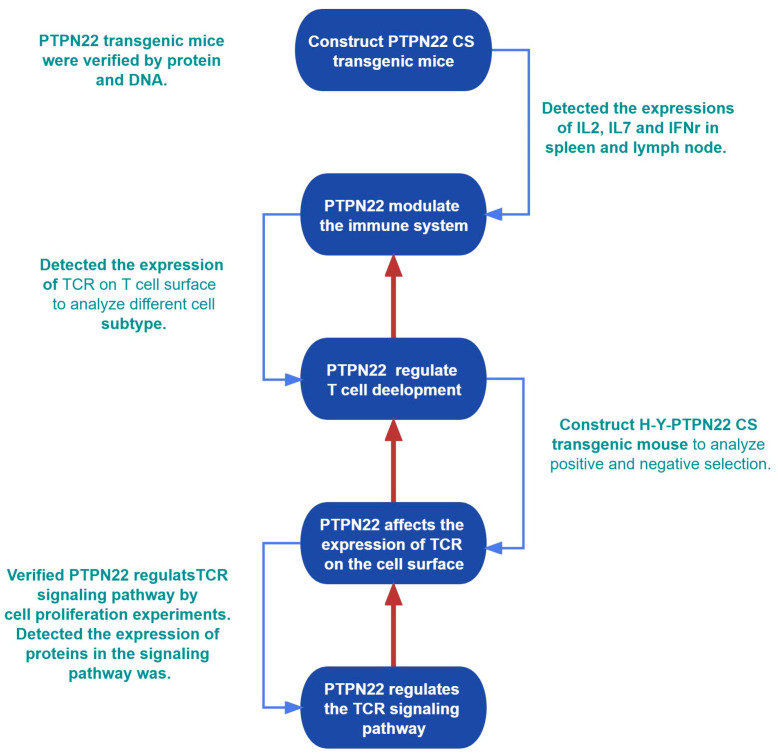
Study the regulation of PTPN22 CS enzyme activity on the development of mouse thymocytes.

## Figures and Tables

**Figure 1 ijms-24-14505-f001:**
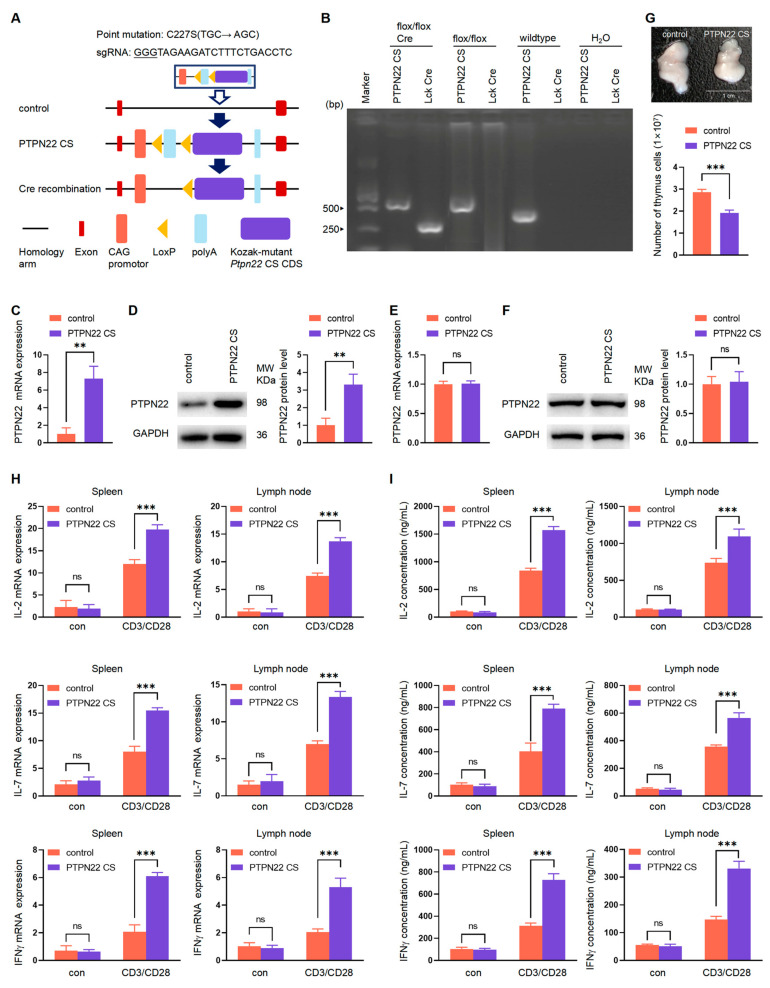
PTPN22 CS transgenic mice were identified in gene and protein. (**A**) PTPN22 CS transgenic mice production process. (**B**) The expression of flox and Lck Cre in mouse thymocytes was detected using PCR. Flox/flox+cre was the PTPN22 CS transgenic mouse that was successfully produced. (**C**) RT-PCR was used to detect the expression of PTPN22 in mouse thymocytes. (**D**) Western blot was used to analyze the expression of PTPN22 in mouse thymocytes. (**E**,**F**) The mRNA and protein expression of PTPN22 in the spleen were detected using RT-PCR and Western blot. The histograms showed the ratio of mRNA or protein. (**G**) The number of thymocytes was detected by flow cytometry. The histograms showed the number of PTPN22 CS or control mouse thymocytes. (**H**) Detected the cytokines expression in the spleen and lymph nodes using RT-PCR. (**I**) Detected the cytokines expression in the spleen and lymph nodes by ELISA. The experiments were repeated more than three times. The values were shown as mean ± SEM, *n* = 10 in (**C**–**F**,**H**,**I**), *n* = 20 in (**G**). ** *p* < 0.01, *** *p* < 0.001. ns, not significant.

**Figure 2 ijms-24-14505-f002:**
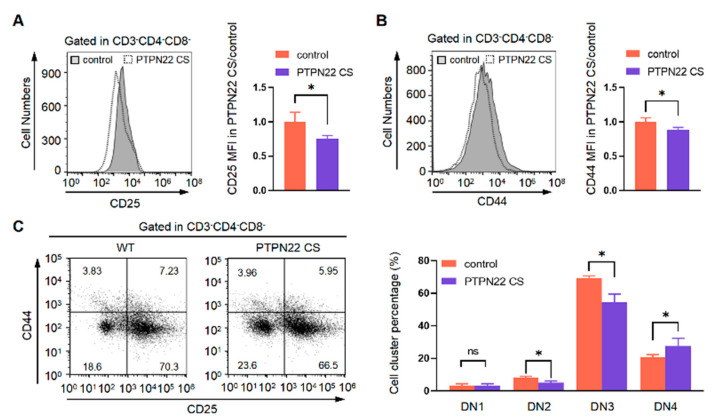
Expressions of CD25 and CD44 were detected on the surface of the thymus cell. (**A**,**B**) The expressions of CD25 and CD44 on the thymus cell surface were detected by flow cytometry. The mouse thymus cells were labeled with CD3, CD4, and CD8. During the flow cytometry analysis, CD3^−^CD4^−^CD8^−^ cells were collected through the setup gate. The histograms are plotted according to the MFI. (**C**) Flow cytometry was used to analyze the proportion of DN1, DN2, DN3, and DN4 cells. The histograms were based on the proportion of CD25^−^CD44^+^, CD25^+^CD44^+^, CD25^+^CD44^−^, and CD25^−^CD44^−^ cells. The experiments were repeated more than three times. The values were shown as mean ± SEM, *n* = 10. * *p* < 0.05. ns, not significant.

**Figure 3 ijms-24-14505-f003:**
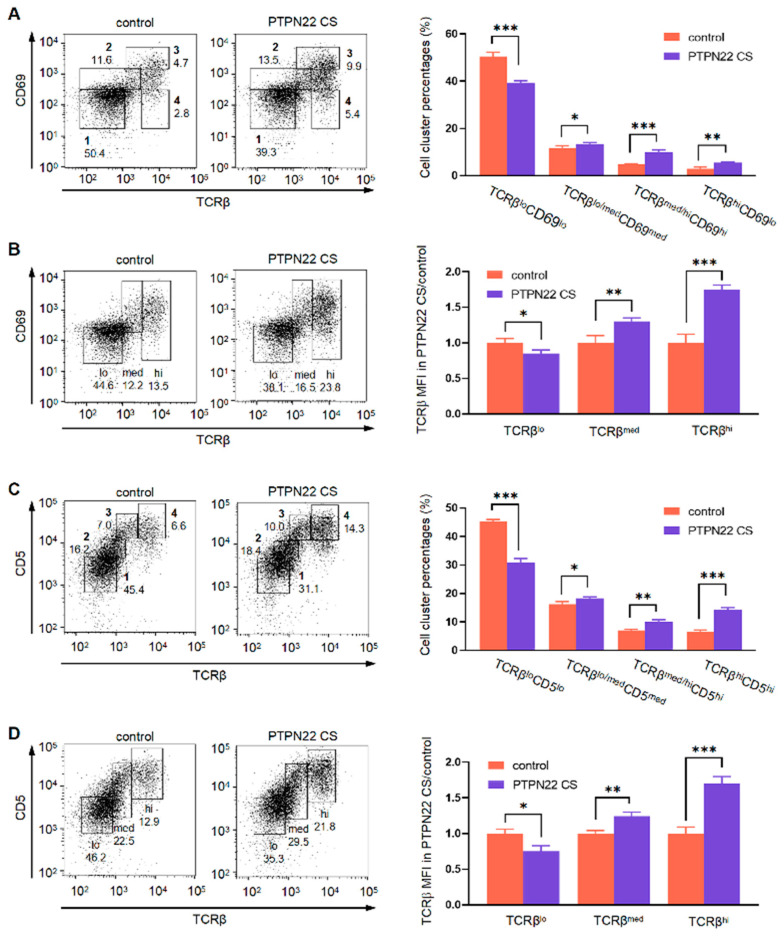
Detected the expression of TCRβ, CD69, and CD5 on the T cell surface. (**A**,**B**) Flow cytometry analyses of the expression of TCRβ and CD69 on the PTPN22 CS and control mouse T cell surface. The thymocytes were taken from PTPN22 CS or control mice at 4–6 weeks old. (**C**,**D**) Flow cytometry analyses of the expression of TCRβ and CD5 on the PTPN22 CS and control mouse T cell surface. The thymocytes were taken from PTPN22 CS or control mice at 4–6 weeks old. The experiments were repeated more than three times. The values were shown as mean ± SEM, *n* = 10. * *p* < 0.05, ** *p* < 0.01, *** *p* < 0.001.

**Figure 4 ijms-24-14505-f004:**
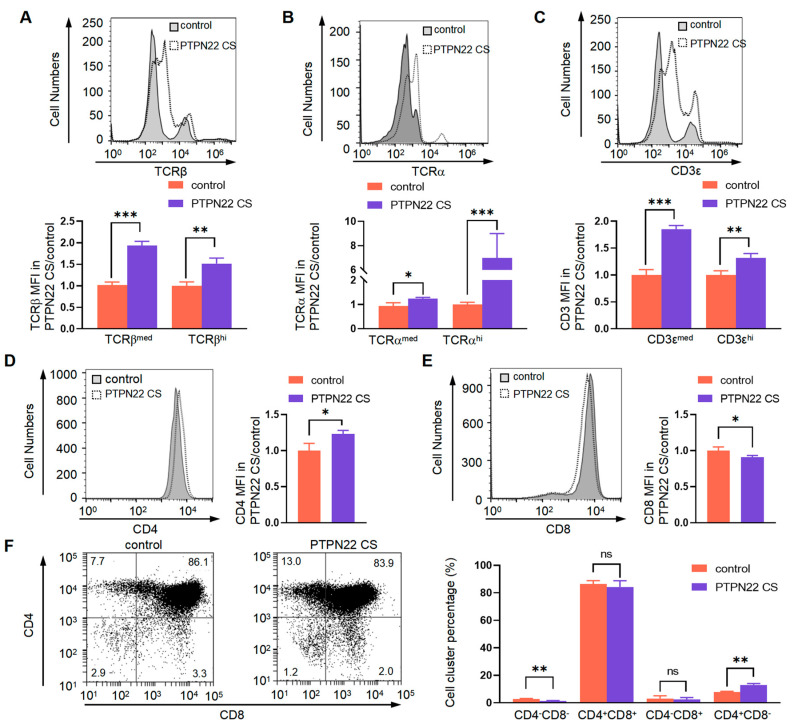
Detected the expression of TCRβ, CD3, CD4, and CD8 on the T cell surface. (**A**–**C**) Analyzed the expression of TCRβ^med^, TCRβ^hi^, TCRα^med^, TCRα^hi^, CD3^med^, and CD3^hi^ on the T cell surface using flow cytometry. The histogram showed the MFI of TCRβ^med^, TCRβ^hi^, TCRα^med^, TCRα^hi^, CD3^med^, and CD3^hi^ of PTPN22 CS and control mouse T cells. (**D**,**E**) Analyzed the expression of CD4 and CD8 on the T cell surface using flow cytometry. The histograms showed the MFI ratio of CD4 and CD8 for PTPN22 CS and control mouse T cells. (**F**) Analyzed the expression of CD4 and CD8 on the T cell surface using flow cytometry. The histograms showed the proportion of cell clusters. The experiments were repeated more than three times. The values were shown as mean ± SEM, *n* = 10. * *p* < 0.05, ** *p* < 0.01, *** *p* < 0.001. ns, not significant.

**Figure 5 ijms-24-14505-f005:**
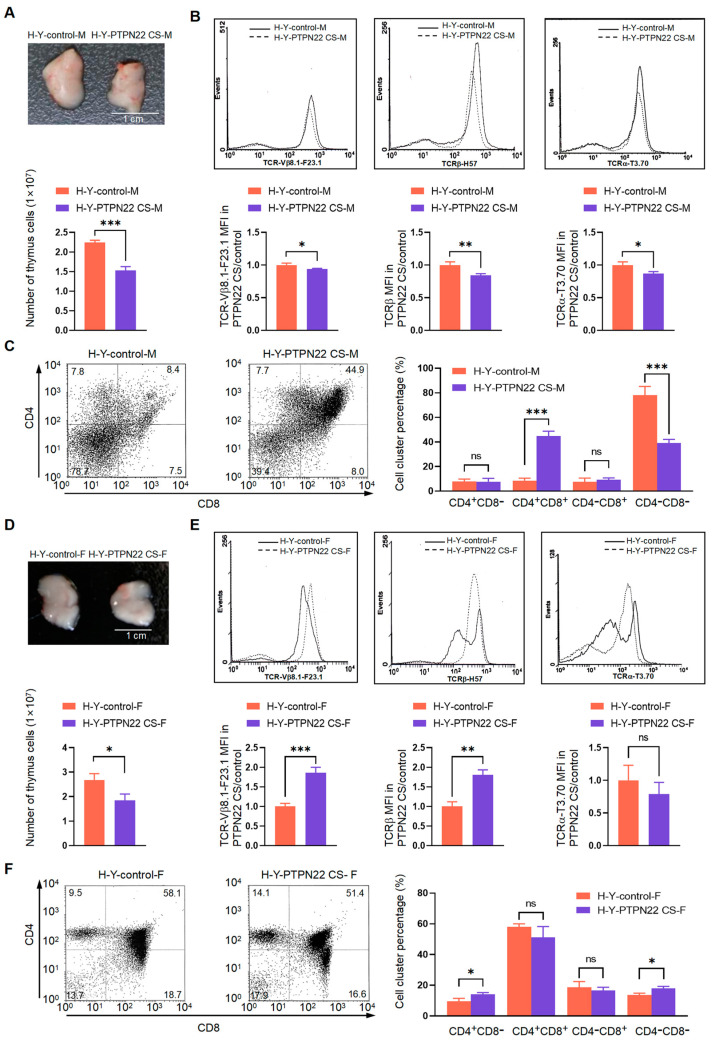
Positive and negative selections were analyzed using flow cytometry. (**A**) Detected the thymus cell number by flow cytometry. Thymus was obtained from H-Y-PTPN22 CS-M or H-Y-control-M mice aged 4–6 weeks. (**B**) Detected the TCRβ and TCRα expression on the T cell surface. (**C**) Flow cytometry analyzed the CD4 and CD8 expression of H-Y-control-M or H-Y-PTPN22 CS-M mice. (**D**) Detected the thymus cell number using flow cytometry. Thymus was obtained from H-Y-PTPN22 CS-F or H-Y-control-F mice aged 4–6 weeks. (**E**) Detected the TCRβ and TCRα expression on H-Y-control-F or H-Y-PTPN22 CS-F mouse T cell surface. (**F**) Flow cytometry analyzed the CD4 and CD8 expression of H-Y-control-F or H-Y-PTPN22 CS-F mice. The experiments were repeated more than three times. The values were shown as mean ± SEM, *n* = 10. * *p* < 0.05, ** *p* < 0.01, *** *p* < 0.001. ns, not significant.

**Figure 6 ijms-24-14505-f006:**
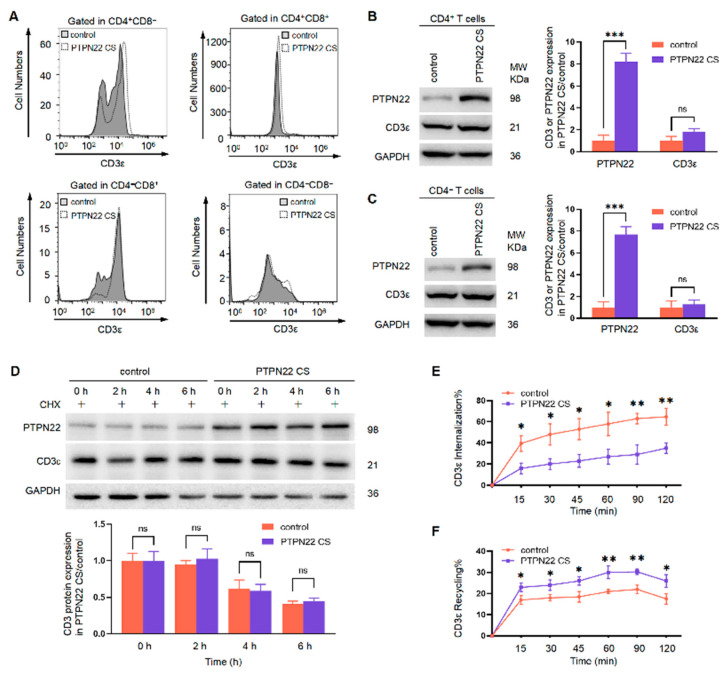
PTPN22 CS regulated TCR internalization and recycling. (**A**) Detected the expression of CD3 on CD4^−^CD8^−^, CD4^+^CD8^−^, CD4^−^CD8^+^, and CD4^+^CD8^+^ cell surfaces using flow cytometry. (**B**,**C**) Detected the expression of CD3 in CD4^+^ or CD4^−^ T cells. CD4^+^ or CD4^−^ T cells were sorted using magnetic beads. The expression of CD3 was detected using Western blot (**D**) Analyzed the degradation rate of CD3 using Western blot. CHX was added to the cell to block protein synthesis. The CD3 expression was detected at different time points. (**E**,**F**) Analyzed the internalization and recycling by flow cytometry. Fluorescent antibodies are labeled on the cell surface. Calculations were performed according to different formulations to analyze internalization and recycling. The experiments were repeated more than three times. The values were shown as mean ± SEM, *n* = 10. * *p* < 0.05, ** *p* < 0.01, *** *p* < 0.001. ns, not significant.

**Figure 7 ijms-24-14505-f007:**
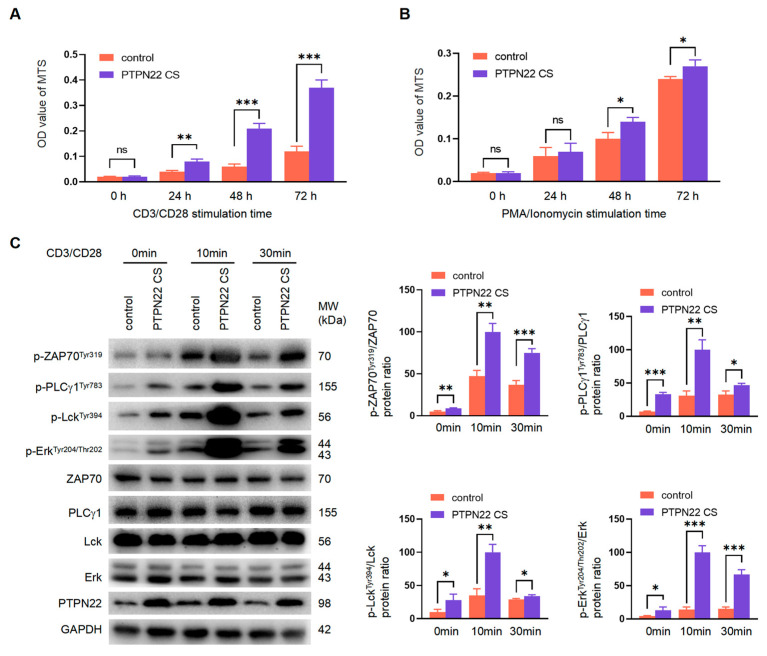
PTPN22 CS affected T cell activation by regulating the TCR signaling pathway. (**A**,**B**) Detected the proliferation of thymocytes. Stimulated the cells by CD3/CD28 or PMA + Ionomycin. Detected the proliferation rate by MTS. (**C**) Detected the phosphorylation level in PTPN22 CS or control mouse thymocytes. The cells were stimulated with CD3 (5 μg/mL)/CD28 (1 μg/mL) for 0, 10, and 30 min. Detected p-ZAP70, p-PLCγ1 p-Lck, and p-Erk using Western blot. The histogram shows the ratio of protein phosphorylation to the corresponding protein. The experiments were repeated more than three times. The values were shown as mean ± SEM, *n* = 10. * *p* < 0.05, ** *p* < 0.01, *** *p* < 0.001. ns, not significant.

## Data Availability

Data is contained within the article or Appendix A.

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
