# Peer review of "The Tyrosine Phosphatase Activity of PTPN22 Is Involved in T Cell Development via the Regulation of TCR Expression"

_ijms, 2023, doi:10.3390/ijms241914505_

Round 1

Reviewer 1 Report

The study of Bin Bai and colleagues investigates the role of  the protein tyrosine phosphatase PTPN22 in the T cells development. PTPN22 inhibits T cell activation by dephosphorylating some essential proteins in the TCR-mediated signaling pathway. PTPN22 has been found to interact with the SH3 domain of C-terminal Src kinase (Csk) to dephosphorylate Lck Y394 and inhibit the activity of Src kinases. Until now, studies of the biological function of PTPN22 have relied primarily on  two mouse models. PTPN22 R619W transgenic mice in which PTPN22 cannot interact with  Csk, resulting in autoimmune diseases, and PTPN22 knockout mice in which there was no statistically significant disparity observed in thymocyte numbers or subsets until they reached the age of six months.

Authors produced PTPN22 CS transgenic mice in which the tyrosine phosphatase activity of PTPN22 is suppressed, in order to investigate the impact of PTPN22 enzyme activity on the initial development of the immune system in mice.The number of thymocytes in the PTPN22 CS mice was significantly reduced. PTPN22 CS facilitated the positive and negative selection of developing thymocytes, increased the expression of TCRε-CD3 complex on thymus cell surface, and regulated their internalization  and recycling.   When mature T cells migrate to spleen and lymph nodes, it was found that the expression of cytokines in spleen and lymph nodes was significantly changed. 

Authors conclude that their  study demonstrates that PTPN22 plays a crucial role in T cell development and provides a theoretical basis for the immune system construction.

 The study is interesting and reveals a novel role of PTPN22 in T cell development. There is no role of PTPN22 in the development  of T cells Æ´δ positive in the thymus? Please, comment

Authors should address the following issues:

 1.      Figure 1: It should be corrected the letter of the second panel. It is B and not A

2.     2.  Figure 2. No values shown in the graph express a significant difference of ** p< 0.01. Please, correct

3.      Line 178 “In particular, PTPN22 CS reduced the proportion of DN cells (Fig. 4F”). But at lines 134-137:” Both DN2 and DN3 cell proportions were reduced in PTPN22 CS mouse, while the proportion of DN4 cells was increased significantly (Fig. 2C). These results suggested that PTPN22 CS promotes the development of DN cells. “ It is not clear. Please, explain better.

4.      Figure 4B - In the graph showing TCRα MFI, MFI of TCRαmed do not seem significantly different between control and PTPN22 CS, even if it is indicated *p<0.05. Please, control the original values.

5.      Line 218: “the proportion of SP cells were higher (Fig. 5E-F)” Only SP CD4+ are higher. Values of SP CD8+ are not significantly different between control and H-Y-PTPN22 CS female

6.      Lines 103, 322, 346 “metastasized to peripheral organs”…….The term metastasize is usually used to indicate cancer cells that spread to other part of the body and cause tumor to grow. You should use “emigrate from the thymus, travel to spleen and other organs, enter blood stream  and etc…….”

7.      References section : ref. 20 and ref. 33 number of pages is missed; ref. 25 the year of publication is missed; ref.15 number of vol. and pages are missed; ref. 6, write the names of the author in full; ref. 41,  the name of the journal should be abbreviated

8.      Extensive English language is required. Some sentences difficult to understand

  Extensive English language is required. Some sentences are not well constructed.

Reviewer 2 Report

The article "The Tyrosine Phosphatase Activity of PTPN22 Is Involved in the Development of T Cells by Regulating TCR Expression", submitted for review, addresses an important problem of immunoregulation. The topic taken up by the authors seems important and timely, but the manuscript needs improvement.

1. The article should be prepared according to the "Instruction for Authors" (including References).

2. Please explain all abbreviations used in the text of the paper.

3. The authors should state the aim of the study.

4. Please draw up a study scheme and state the size of the study sample.

5. In the discussion, the authors should compare their results with the results of other studies. Furthermore, they should explain in detail the relevance of the obtained results for clinical practice.

6. Please complete the "Conclusions" section. The authors should set a direction for further research.
